# Role of *Cnot6l* in maternal mRNA turnover

Filip Horvat[1,2,]*, Helena Fulka[1,3,]*, Radek Jankele[1,]*, Radek Malik[1], Ma Jun[4,5], Katerina Solcova[1], Radislav Sedlacek[6], Kristian Vlahovicek[2], Richard M Schultz[4,5], Petr Svoboda[1]

**Removal of poly(A) tail is an important mechanism controlling eukaryotic mRNA turnover. The major eukaryotic deadenylase complex CCR4-NOT contains two deadenylase components, CCR4 and CAF1, for which mammalian CCR4 is encoded by *Cnot6* or *Cnot6l* paralogs. We show that *Cnot6l* apparently supplies the majority of CCR4 in the maternal CCR4-NOT in mouse, hamster, and bovine oocytes. Deletion of *Cnot6l* yielded viable mice, but *Cnot6l*$^{-/-}$ females exhibited ~40% smaller litter size. The main onset of the phenotype was post-zygotic: fertilized *Cnot6l*$^{-/-}$ eggs developed slower and arrested more frequently than *Cnot6l*$^{+/-}$ eggs, suggesting that maternal CNOT6L is necessary for accurate oocyte-to-embryo transition. Transcriptome analysis revealed major transcriptome changes in *Cnot6l*$^{-/-}$ ovulated eggs and one-cell zygotes. In contrast, minimal transcriptome changes in preovulatory *Cnot6l*$^{-/-}$ oocytes were consistent with reported *Cnot6l* mRNA dormancy. A minimal overlap between transcripts sensitive to decapping inhibition and *Cnot6l* loss suggests that decapping and CNOT6L-mediated deadenylation selectively target distinct subsets of mRNAs during oocyte-to-embryo transition in mouse.**

## Introduction

During the oocyte-to-embryo transition (OET), maternal mRNAs deposited in the oocyte are gradually replaced by zygotic mRNAs. Consequently, control of mRNA stability is a principal mechanism assuring correct gene expression reprogramming at the beginning of development. Maternal mRNA degradation during mouse OET occurs in several distinct waves (reviewed in detail in references 1 and 2). Control of mRNA stability involves various mechanisms target, many employing protein interaction with the 3′ untranslated region, that ultimately target the terminal 5′ cap and 3′ poly(A) tail structures (reviewed in reference 3). The main mammalian mRNA decay pathway involves deadenylation

coupled with decapping (4). Eukaryotic cells employ three main deadenylases: CCR4-NOT (carbon catabolite repression 4–negative on TATA-less) complex, PAN2/3 complex, and PARN, which differ in sensitivity to cap structure, poly(A) tail length, and poly(A)-binding protein (PABP) (reviewed in references 5 and 6). Cytoplasmic mRNA decay in mammalian cells initiates at the 3′ end and involves sequential deadenylation, first by PAN2/3 followed by CCR4–NOT (4, 7). However, recent data suggest that CCR4–NOT–mediated deadenylation is the main pathway in general mRNA turnover (8).

The multiprotein CCR4–NOT complex (reviewed in references 9 and 10) was first identified in *Saccharomyces cerevisiae* as a gene regulating glucose-repressible alcohol dehydrogenase 2. The mammalian CCR4–NOT complex (Fig 1A) is composed of a docking platform (CNOT1) that binds regulatory components (CNOT2, CNOT3, CNOT4, CNOT9, CNOT10, and CNOT11) and two deadenylase components equivalent to yeast's CAF1 and CCR4 deadenylases. CAF and CCR4 differ with respect to their relationship with the PABP; CCR4 can degrade poly(A) bound with PABP, whereas CAF1 degrades free poly(A) (8). Mammals use two paralogs of CAF1 (CNOT7 and CNOT8) and two of CCR4 (CNOT6 and CNOT6L). Thus, a CCR4–NOT complex carries one of four possible combinations of CAF1 and CCR4 homologs. However, the significance of different CCR4–NOT variants remains unclear.

The CCR4–NOT complex can be recruited to mRNA by different BTG/Tob proteins, selective RNA-binding proteins such as tristetraprolin, or upon miRNA binding through TNRC6A-C proteins (reviewed in reference 11). Although miRNA-mediated mRNA degradation is insignificant (12), BTG4 plays a major role in maternal mRNA degradation (13). Another mechanism of selective mRNA targeting by CCR4–NOT is direct recruitment of the complex through YTHDF2, which binds the N6 adenosine (m$^6$A) RNA modification (14). YTHDF2 was linked to selective elimination of maternal mRNAs during oocyte maturation (15).

Maternal mRNAs in mouse oocytes are unusually stable during the growth phase before oocyte maturation, which is accompanied

[1]Institute of Molecular Genetics of the Czech Academy of Sciences, Prague, Czech Republic  [2]Bioinformatics Group, Division of Molecular Biology, Department of Biology, Faculty of Science, University of Zagreb, Zagreb, Croatia  [3]Institute of Animal Science, Prague, Czech Republic  [4]Department of Biology, University of Pennsylvania, Philadelphia, PA, USA  [5]Department of Anatomy, Physiology, and Cell Biology, School of Veterinary Medicine, University of California, Davis, CA, USA  [6]Czech Centre for Phenogenomics and Laboratory of Transgenic Models of Diseases, Institute of Molecular Genetics of the Czech Academy of Sciences, v. v. i., Vestec, Czech Republic

Correspondence: svobodap@img.cas.cz
Radek Jankele's present address is Swiss Institute for Experimental Cancer Research (ISREC), School of Live Sciences, Swiss Federal Institute of Technology Lausanne (EPFL), Lausanne, Switzerland.
*Filip Horvat, Helena Fulka, and Radek Jankele contributed equally to this work.

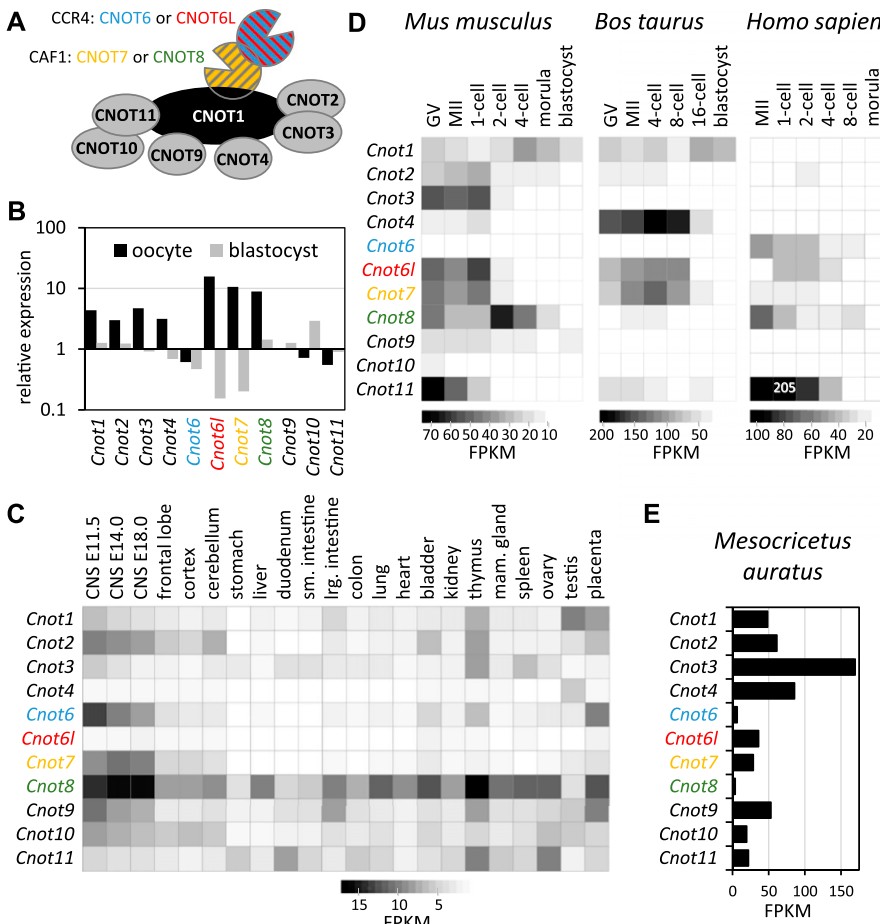

**Figure 1. High maternal expression of *Cnot6l*, an active component of the CCR4–NOT deadenylase complex.**

**(A)** Schematic depiction of the mammalian CCR4–NOT deadenylase complex. The organization of the complex was compiled from the literature (9, 10, 41). Each CCR4–NOT complex contains two active deadenylase proteins, which can make four possible combinations: CNOT7+CNOT6, CNOT7+CNOT6L, CNOT8+CNOT6, and CNOT8+CNOT6L. CNOT6, CNOT6L, CNOT7, and CNOT8 are color-coded for easier navigation in the other panels. **(B)** *Cnot6l* transcript is highly enriched in the oocyte relative to somatic cells, whereas its alternating paralog *Cnot6* is relatively depleted. The graph shows an expression ratio of CCR4–NOT complex components in oocytes and blastocysts relative to median expression values in somatic tissues calculated from the BioGPS GNF1M.gcrma tissue expression dataset (18) where expression in 61 mouse tissues was set to one. Expression of *Cnot* genes in somatic tissues **(C)**, during OET in mice, cattle, and humans **(D)**, and in hamster oocytes **(E)**. Heatmaps and the graph show fragments per kilobase per million (FPKMs). Expression data from 22 tissues were selected from the ENCODE polyA RNA-seq mouse tissue panel (GSE49417) (42), and expression analysis of *Cnot* genes in oocytes and early embryos is based on published datasets from indicated species (19, 20, 21, 22, 35).

with a transition from mRNA stability to instability (reviewed in reference 2). This transition also involves recruitment of dormant maternal mRNAs that were accumulated but not (or poorly) translated during the growth phase. Some dormant mRNAs encode components of mRNA degradation pathways (13, 16, 17) and include DCP1A and DCP2, which are critical components of the decapping complex (16). Inhibiting the maturation-associated increase in DCP1A and DCP2 results in stabilizing a subset of maternal mRNAs that are normally degraded and affects zygotic genome activation (16). Dormancy was also shown for BTG4 (13) and components of deadenylase complexes: PAN2 for the PAN2/3 complex and CNOT7 and CNOT6L for the CCR4–NOT complex (17).

Here, we report an analysis of *Cnot6l* function during OET in mice. Transcripts encoding the CCR4–NOT complex are relatively more abundant in mouse oocytes than in the blastocyst or in somatic tissues. *Cnot6l* expression apparently supplies the majority of the CCR4 component of the maternal CCR4–NOT complex in mouse, hamster, and bovine, but not human, oocytes. Mice lacking *Cnot6l* are viable and fertile. However, zygotes arising from *Cnot6l$^{-/-}$* eggs develop slower and more likely developmentally arrest than zygotes from heterozygous eggs. Correspondingly, *Cnot6l$^{-/-}$* females exhibit ~40% lower fertility. Consistent with the previous report that *Cnot6l* is a dormant maternal mRNA (17), transcriptome analysis revealed minimal transcriptome changes in *Cnot6l$^{-/-}$* germinal vesicle-intact (GV) oocytes. Nevertheless, there is a subset of maternal mRNAs that are stabilized during oocyte maturation and after fertilization, suggesting that CNOT6L primarily acts in maternal mRNA degradation during oocyte maturation and in zygotes.

# Results and Discussion

### Mammalian CCR4 paralog CNOT6L is highly expressed in oocytes

Several components of the CCR4–NOT complex have a higher relative expression in oocytes in the gcRMA mouse set in the GNF Symatlas database (18) (Fig 1B). Of the four genes encoding active deadenylase components of the CCR4–NOT complex, the CCR4 paralog *Cnot6* showed slightly lower expression when compared with a panel of somatic tissues, whereas transcript abundance of the *Cnot6l* paralog appeared highly enriched in oocytes. These data suggested that CNOT6L could be the main CCR4 deadenylase component during OET (Fig 1B). In contrast, *Cnot6* appeared to be highly expressed in many somatic tissues, particularly in embryonic neuronal tissues (Fig 1C).

For further insight into the expression of the CCR4–NOT complex during OET, we analyzed transcript levels of individual components in RNA-sequencing (RNA-seq) data from OET in mouse (19, 20), cow

(21), and human (22) oocytes (Fig 1D). In mouse oocytes, *Cnot6l* mRNA level was approximately 10 times higher than *Cnot6*, which is expressed during oocyte growth and is apparently not a dormant maternal mRNA (23). High *Cnot6l* and low *Cnot6* expression during OET was also observed in cow but not in humans, where the level of *Cnot6* transcript was higher in metaphase II (MII) eggs. A subsequent equalization of *Cnot6* and *Cnot6l* expression in human zygotes (Fig 1D) could be a consequence of cytoplasmic polyadenylation of dormant *Cnot6l* mRNA, which can manifest as an apparent increase in mRNA level in poly(A) RNA-seq data (24). High *Cnot6l* and low *Cnot6* expression was also found in GV oocytes of golden hamster, suggesting that this difference in expression is a conserved feature in rodents (Fig 1E). Interestingly, CAF1 paralogs *Cnot7* and *Cnot8* showed more variable patterns, including equal expression of both paralogs in mouse, dominating *Cnot7* in bovine and golden hamster, and dominating *Cnot8* paralog expression in human oocytes.

### *Cnot6l* knockout is viable but exhibits reduced fertility

Given the dominant maternal expression of *Cnot6l* relative to its paralog *Cnot6*, we decided to examine the role of *Cnot6l* in mice using a TAL effector nuclease (TALEN)–mediated knockout. We designed two TALEN pairs, which would induce ~31.3-kb deletion, affecting exons 5–12 (Fig 2A). This deletion, which would eliminate the entire CC4b deadenylase domain and a part of the upstream leucine-rich repeat region, was expected to genetically eliminate the CNOT6L protein. We obtained two founder animals carrying two very similar deletion alleles (Figs 2B and S1). Interestingly, the second allele (Cnot6L-del5-12b) contained a 25-bp insert apparently derived from mitochondrial DNA.

Male and female *Cnot6l*$^{-/-}$ mice appeared normal and were fertile. *Cnot6l* is thus a nonessential gene. Breeding heterozygotes or *Cnot6l*$^{-/-}$ males with *Cnot6l*$^{+/-}$ females yielded on average 6.9 ± 1.6 and 6.2 ± 1.9 pups per litter, respectively (Table 1), which is consistent with the reported C57BL/6 litter size of 6.2 ± 0.2 (25). We typically observe six to eight animals per litter in the C57BL/6 strain used to produce mouse models in our facility (20). Analysis of the *Cnot6l*$^{-/-}$ breeding data showed that an average litter size of approximately four pups of *Cnot6l*$^{-/-}$ females mated with *Cnot6l*$^{+/+}$, *Cnot6l*$^{+/-}$, or *Cnot6l*$^{-/-}$ males (Table 1). Reduced litter sizes of *Cnot6l*$^{-/-}$ females that mated with males of any of the three genotypes were statistically significant ($P < 0.01$, two-tailed $t$ test) when compared with the litter size of *Cnot6l*$^{+/-}$ animals. The breeding data thus indicated a maternal-effect phenotype and showed no evidence for a significant role of zygotic and embryonic expression of *Cnot6l*.

Superovulated knockout females yielded on average the same number of MII eggs as heterozygote littermates and wild-type C57BL/6 (31.5 versus 32.3 versus 28.5, respectively; n = 6), suggesting that the reduction of litter size occurs during fertilization or after fertilization. Accordingly, we analyzed early development of zygotes derived from *Cnot6l*$^{-/-}$ and *Cnot6l*$^{+/-}$ eggs fertilized in vitro with wild-type sperm and observed a small but significant reduction of fertilization efficiency of *Cnot6l*$^{-/-}$ versus *Cnot6l*$^{+/-}$ eggs (85 versus 99%; 133/155 *Cnot6l*$^{-/-}$ versus 203/205 *Cnot6l*$^{+/-}$ eggs that formed zygotes; Fisher's test $P$-value < 0.001). Analysis of the cleavage times of embryos using a PrimoVision time-lapse system revealed a small but significant delay in early development that could contribute to the reduced litter size (Fig 2C). Importantly, although there was no stage-specific arrest of development for *Cnot6l*$^{-/-}$-fertilized eggs, they were two times more likely to fail to reach the blastocyst stage than their *Cnot6l*$^{+/-}$-derived counterparts

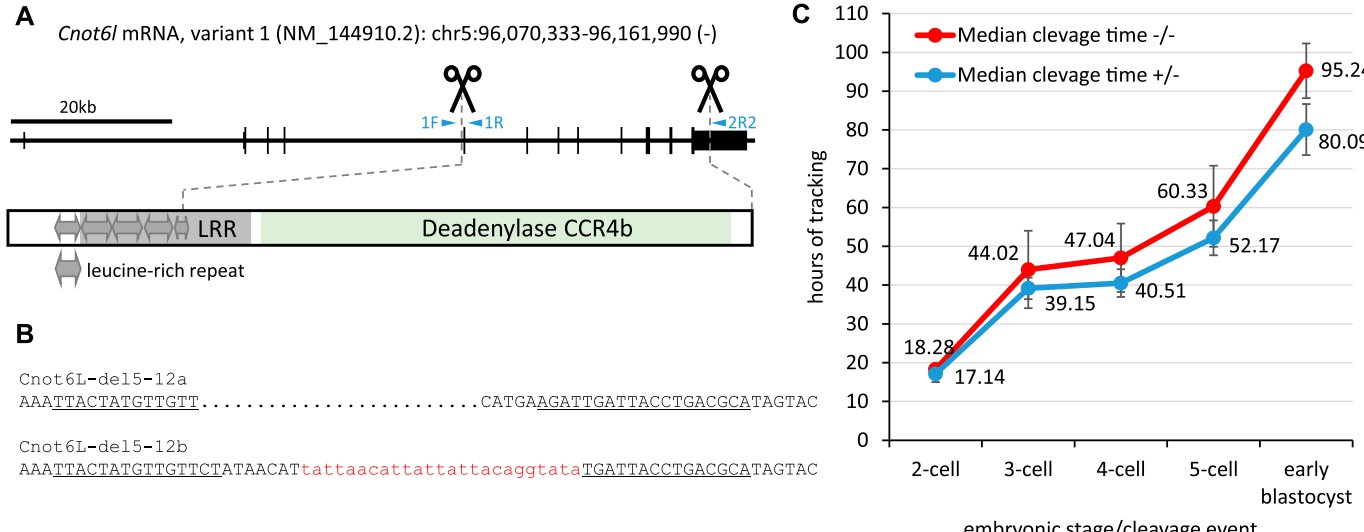

**Figure 2. TALEN-mediated knockout of *Cnot6l* gene in mice.**
**(A)** A scheme of *Cnot6l* gene depicting the position of the deletion in the genomic DNA and the corresponding part of the CNOT6L protein. Protein domains were mapped using the Conserved Domain Database search (43). **(B)** Sequences of two alleles identified in F$_0$ animals, which carry ~31.3-kb deletions of exons 4–11. Underlined sequences indicate TALEN cognate sequences. A short fragment of mitochondrial DNA integrated into the deleted locus is visualized in lowercase red font. **(C)** Zygotes from in vitro–fertilized *Cnot6l*$^{-/-}$ eggs develop significantly slower ($t$ test, $P$-value < 0.001 for all stages) than zygotes developing from heterozygous eggs. Error bar = SD. In total, 95 and 133 zygotes produced from *Cnot6l*$^{-/-}$ (−/−) and *Cnot6l*$^{+/-}$ (+/−), respectively, were analyzed using the PrimoVision time-lapse system. Numbers indicate hours from the point the zygotes were placed into the tracking system (5 h after mixing sperm with cumulus oocyte complex), which automatically detects the first four cleavage events and formation of early blastocysts.

**Table 1.  Breeding performance of *Cnot6l* mutants.**

| F × M | +/+ | +/− | −/− | n.d. | M | F | Litters | Litter size (±SD) |
|---|---|---|---|---|---|---|---|---|
| +/− × +/− | 26 | 30 | 18 | 2 | 39 | 37 | 11 | 6.9 (±1.6) |
| +/− × −/− | 0 | 22 | 32 | 2 | 28 | 28 | 9 | 6.2 (±1.9) |
| −/− × +/+ | 0 | 48 | 0 | 1 | 27 | 22 | 13 | 3.8 (±1.6) |
| −/− × +/− | 0 | 4 | 11 | 13 | 9 | 19 | 7 | 4.0 (±1.4) |
| −/− × −/− | 0 | 0 | 21 | 0 | 9 | 12 | 5 | 4.2 (±1.8) |

(38/95 [40.0%] *Cnot6l*$^{-/-}$ versus 26/133 [19.6%] *Cnot6l*$^{+/-}$-fertilized eggs failed to develop to the blastocyst). This observation suggested reduced developmental competence of *Cnot6l*$^{-/-}$ zygotes and likely accounted for the reduced litter size (Table 1).

### Small but significant transcriptome changes in Cnot6l$^{-/-}$ oocytes and zygotes

To explore the impact of *Cnot6l* loss on the transcriptome during OET, we performed RNA-seq analysis of GV oocytes, MII eggs, and one-cell zygotes. All replicates showed good reproducibility (Figs S2 and S3). RNA-seq data showed minimal levels of transcripts arising from the deleted *Cnot6l* allele (Figs 3A and S4). In addition, we did not observe any compensatory change in *Cnot6* mRNA expression (Fig 3B).

Transcription from the deleted locus yielded low levels of aberrant transcripts where the splice donor of the third coding exon of *Cnot6l* was spliced with five different downstream splice acceptor sites in the adjacent intron or downstream of the last *Cnot6l* exon (Fig S4). In all cases, exons spliced with the third coding exon contained stop codons and, thus, all transcripts generated from the deleted locus would encode the truncated CNOT6L protein composed of the N-terminal leucine repeats. It is unlikely that such isoforms would affect fertility as they would also be present in oocytes of *Cnot6l*$^{+/-}$ females, which have normal fertility.

Principal component analysis indicated a small magnitude of changes in knockout samples as the samples clustered primarily by developmental stages (Fig 3C). Analysis of differentially expressed transcripts using DESeq2 package (26) with the default *P*-value cutoff 0.1 showed minimal transcriptome changes in GV oocytes (only four

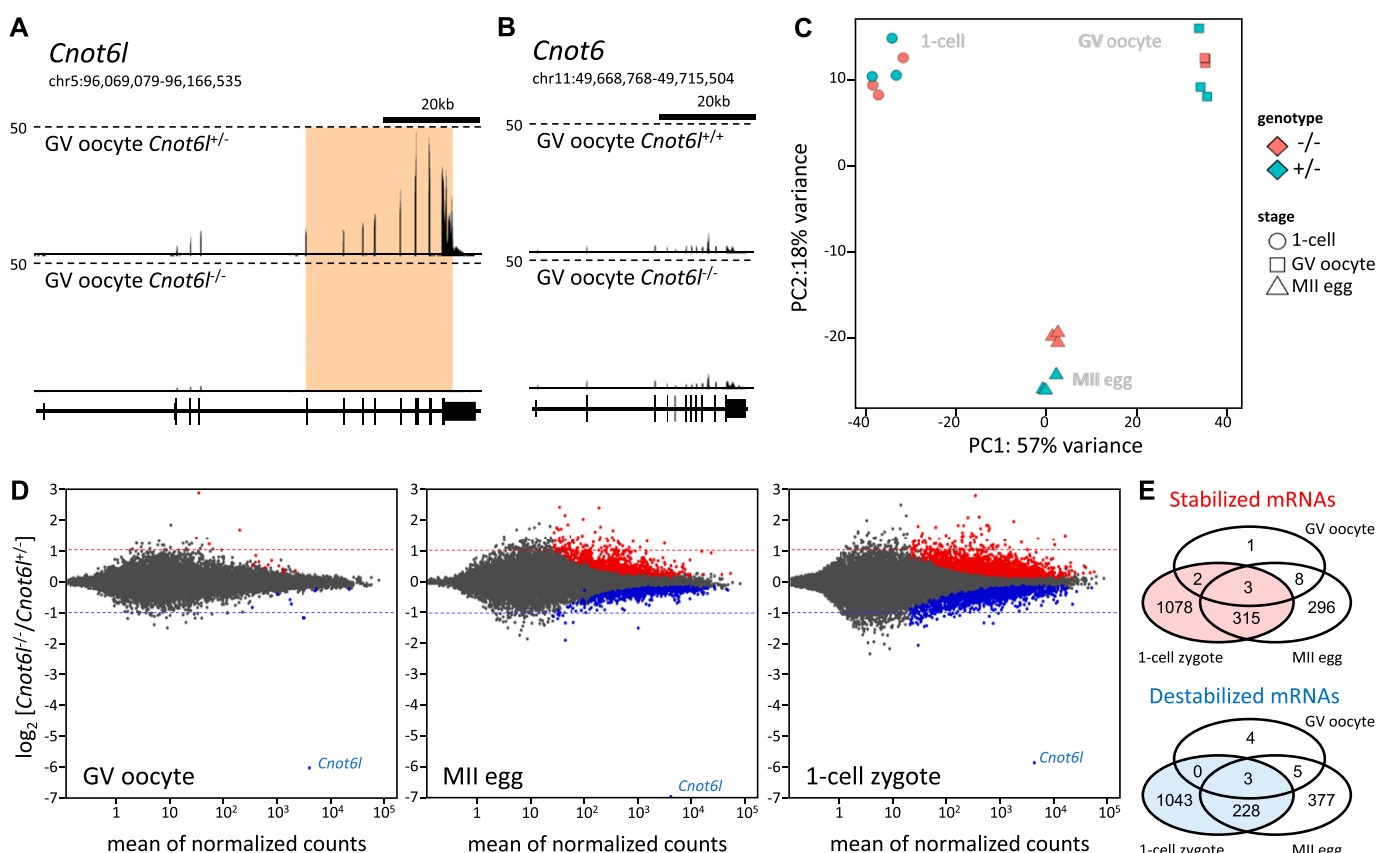

**Figure 3.  Transcriptome changes in *Cnot6l* knockout oocytes and zygotes.**
**(A)** Transcriptional landscape in the *Cnot6l* locus in oocytes from *Cnot6l*$^{+/-}$ and *Cnot6l*$^{-/-}$ animals. Shown is a UCSC genome browser snapshot (36) of the *Cnot6l* locus with expression data from one of the replicates of *Cnot6l*$^{+/-}$ and *Cnot6l*$^{-/-}$ samples. The orange region indicates the region deleted in knockouts. **(B)** Loss of *Cnot6l* expression has no effect on *Cnot6* expression. Shown is a UCSC genome browser snapshot of the *Cnot6* locus from the same samples as in panel (A). **(C)** Principal component analysis of transcriptomes of *Cnot6l*$^{-/-}$ and control oocytes and zygotes. Heterozygous littermates were used as controls in case of GV and one-cell zygotes; age-matched C57BL/6 females were used as controls for MII eggs because there were not enough *Cnot6l*$^{+/-}$ littermates for all control samples. **(D)** Differentially expressed transcripts in *Cnot6l*$^{-/-}$ GV oocytes, MII eggs, and one-cell zygotes. MA plots depict genes with significantly higher (red) or lower (blue) mRNA abundance. Dashed lines depict twofold change for easier navigation. The outlier gene at the bottom of each graph is *Cnot6l*. **(E)** Venn diagrams depicting numbers of genes showing significantly different transcript abundances in *Cnot6l*$^{-/-}$ oocytes and zygotes.

transcripts showing a significant increase in abundance greater than twofold), which indicates that *Cnot6l* is not required for the formation of the maternal transcriptome. There was, however, an apparent progressive transcriptome disturbance in MII eggs and one-cell zygotes (Fig 3D and Table S1–S3). This finding is consistent with the previously reported dormancy of *Cnot6l* (17) and the hypothesis that the reduced litter size of *Cnot6l*$^{-/-}$ females is a maternal-effect phenotype. The magnitude of transcriptome disturbance appears small despite the number of significantly affected genes; if RNA-seq data would be quantified as transcripts per million, higher transcript levels of 622 genes in MII eggs (Fig 3E) would account for 1.28% of the transcriptome.

Interestingly, the numbers of significantly up-regulated and down-regulated mRNAs were comparable (Fig 3E), which was unexpected because transcript stabilization would be the primary expected effect of a deadenylase component loss from the CCR4–NOT complex. It is possible that preventing CNOT6L-mediated deadenylation (hence destabilization) of transcripts from several hundred genes might result in accelerated degradation of other transcripts, noting that we previously observed a similar phenomenon when the maturation-associated increase in DCP1A/DCP2 was inhibited (16). In any case, when the significantly stabilized transcripts were projected onto transcriptome changes during maturation and following fertilization, it was clear that exclusive CNOT6L-dependent destabilization of maternal transcripts concerns only a smaller fraction of maternal mRNAs degraded during OET (Fig 4A).

Relative transcript changes during OET can be problematic to interpret because they may reflect changes in poly(A) tail length and not changes in transcript abundance due to mRNA degradation or transcription (24). Although the Ovation system used for producing RNA-seq libraries uses total RNA as input material, genome-mapped data show that mRNAs are preferentially sequenced and

the sequencing yields a slight bias toward mRNAs with longer poly(A) tails (e.g., *Mos* mRNA, a typical dormant maternal mRNA polyadenylated during meiotic maturation [27], showed an apparent ~17% increased abundance in control wild-type MII eggs relative to GV oocytes).

To examine a potential impact of poly(A) tail length on transcript abundance during meiotic maturation, we used a published poly(A) tail sequencing dataset (28) to generate a plot of the relative change in transcript abundance in MII eggs as a function of poly(A) tail length (Fig 4B). These data showed that transcripts showing relatively increased abundance in *Cnot6l*$^{-/-}$ eggs typically have longer poly(A) tails (60–80 nt). However, when taking into account the distribution of poly(A) lengths in the entire transcriptome, the relative frequency of transcripts with increased abundance in *Cnot6l*$^{-/-}$ eggs was similar for transcripts with poly(A) tails 30–80 nt in length (Fig 4C), suggesting that Fig 4B data only reflect that most maternal transcripts have poly(A) tails 60–80 nt long.

To further resolve the issue of mRNA abundance versus poly(A)-length effects in differentially expressed transcripts in zygotes derived from *Cnot6l*$^{-/-}$ eggs, we used RNA-seq datasets from MII eggs and one-cell zygotes that were generated from directly selected poly(A) and from total RNA without any poly(A) bias (19, 20). These data allow distinguishing between true mRNA degradation, which would be observed in the total RNA data, and deadenylation/polyadenylation, which would manifest in the poly(A) data (Fig 4D). When transcripts showing a significant relative increase in *Cnot6l*$^{-/-}$ zygotes were projected on these data, there was a clear shift to the left on the *x*-axis, consistent with their deadenylation. Furthermore, a fraction of these transcripts also showed apparent degradation as evidenced by their position on the *y*-axis (Fig 4D). Altogether, these data show that maternal *Cnot6l* contributes to maternal mRNA deadenylation and degradation during OET.

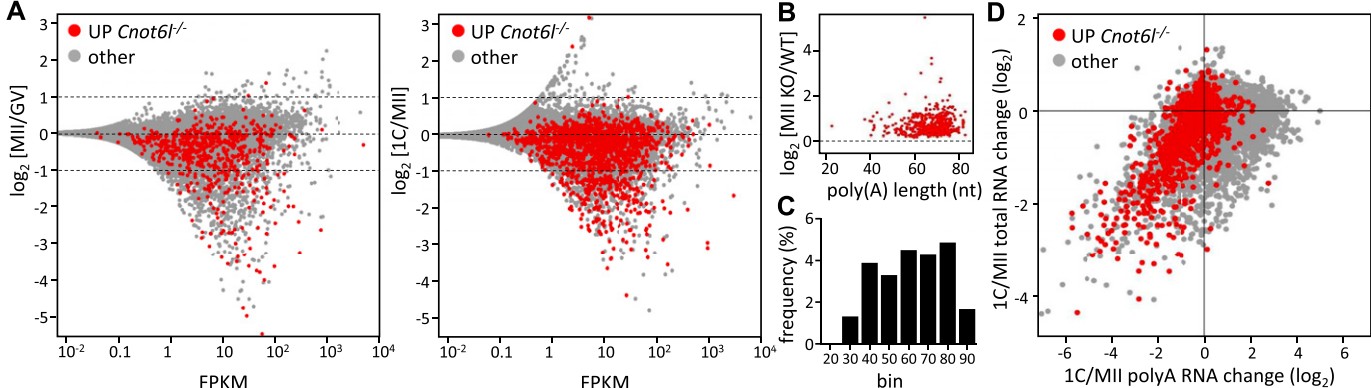

**Figure 4. Transcriptome changes in Cnot6l knockout oocytes and zygotes.**
**(A)** Projection of differentially expressed transcripts in *Cnot6l*$^{-/-}$ MII eggs and zygotes onto transcriptome changes during meiotic maturation and after fertilization. MA plots were constructed from wild-type samples as indicated and shown as red genes with significantly higher mRNA abundance in *Cnot6l*$^{-/-}$ MII eggs (left graph) and zygotes (right graph). **(B)** Analysis of poly(A) tail length of mRNAs significantly increased in *Cnot6l*$^{-/-}$ oocytes during meiosis. The *y*-axis shows the relative up-regulation of maternal mRNAs in *Cnot6l*$^{-/-}$ MII eggs (same genes as those labeled in red in MII eggs in Fig 3D). The *x*-axis depicts poly(A) tail length in GV oocytes taken from the literature (28). **(C)** Relative distribution of mRNAs significantly increased *Cnot6l*$^{-/-}$ oocytes according to poly(A) tail length. RNAs were binned according to the poly(A) tail length into 10-nt bins (0–10, 10–20, 20–30, etc.), and the number of transcripts significantly increased in *Cnot6l*$^{-/-}$ MII eggs was divided with the total number of transcripts in each bin. The *x*-axis numbers represent the upper values of binned poly(A) tail lengths. According to the poly(A) length in GV oocytes, the *y*-axis shows the relative up-regulation of maternal mRNAs in *Cnot6l*$^{-/-}$ MII eggs (corresponding to genes labeled in red in MII eggs in Fig 3D). The *x*-axis depicts the poly(A) tail length in GV oocytes taken from the literature (28). **(D)** Transcriptome changes in *Cnot6l*$^{-/-}$ zygotes are consistent with a role for CNOT6L in deadenylation during OET. Published RNA-seq data for relative poly(A) RNA and total RNA changes (19, 20) were used to construct the plot. In red are shown genes with significantly higher mRNA abundance in *Cnot6l*$^{-/-}$ zygotes. The *y*-axis shows relative changes in total RNA (i.e., RNA degradation), whereas the *x*-axis shows poly(A) RNA changes (i.e., RNA degradation and/or deadenylation).

Finding that the transcriptome changes following *Cnot6l* loss are restricted to a fraction of deadenylated and degraded maternal mRNAs suggests some selectivity of a CNOT6L-containing CCR4–NOT complex in targeting mRNAs. CCR4–NOT complex recruitment to maternal mRNAs through BTG4 does not appear very selective given the large number of affected maternal mRNAs in *Btg4*$^{-/-}$ MII eggs, which includes ~1/3 of the transcripts showing a relative increase in *Cnot6l*$^{-/-}$ MII eggs (Fig 5A). These data also suggest that the CAF1 (CNOT7 and CNOT8) component of the CCR4–NOT complex is probably sufficient for a large part of deadenylation mediated by the CCR4–NOT complex. This suggestion is also consistent with the effect of CNOT7 knockdown in early embryos (17), which is apparently more detrimental for early development than the loss of CNOT6L reported here.

To gain further insight into the potential selectivity of CNOT6L-mediated mRNA deadenylation and degradation, we examined overlaps with transcriptome changes in *Ythdf2* knockout eggs (15) (Fig 5B), in *Tut4/7*$^{-/-}$ oocytes (28) (Fig 5C) and in eggs with suppressed decapping (16) (Fig 5D). In all three cases, transcriptome changes concerned hundreds of transcripts. Accordingly, we assessed whether CNOT6L contributes to selective targeting of m6A-marked maternal mRNAs during meiotic maturation and to what extent mRNAs destabilized through CNOT6L and the decapping complex are mutually exclusive. In all cases, the overlap of transcripts whose relative abundance is increased in MII eggs was minimal (although statistically significant in the case of the *Ythdf2* knockout [Fisher's exact test *P*-value = 4.06e −14]). Furthermore, transcripts regulated by *Ythdf2* and *Tut4/7* were apparently less expressed during meiotic maturation (<10 FPKM; Fig 5B and C) than transcripts targeted by decapping (Fig 5D).

In any case, maternal mRNAs preferentially targeted through decapping are, thus, a distinct group from those stabilized upon

elimination of *Cnot6l*. This difference becomes apparent when these transcripts are visualized in transcriptome data from unfertilized and fertilized eggs resolved according to relative abundance in total RNA and poly(A) RNA-seq (19, 20) (Fig 6). In this display, the *y*-axis corresponds to RNA degradation and the *x*-axis reflects poly(A) changes. Deadenylated and degraded RNAs are found in the lower left quadrant. When transcripts up-regulated in *Cnot6l*$^{-/-}$ MII eggs or up-regulated in MII eggs upon inhibition of decapping are highlighted in this plot, transcripts most sensitive to decapping inhibition seem to be degraded without pronounced deadenylation, unlike transcripts sensitive to *Cnot6l* loss (Fig 6).

A selective function has been proposed for CNOT6-mediated deadenylation of maternal mRNAs. CNOT6 is present in full-grown GV oocytes in cortical foci and regulates deadenylation of mRNAs such as *Orc6* or *Slbp* that were transiently polyadenylated during early meiotic maturation (23). Remarkably, CNOT6 and CNOT6L paralogs are highly similar at the protein level (Fig S5); the major differences concern the five–amino-acid residue longer N terminus of CNOT6L and the five–amino-acid residue insertion in CNOT6 at the end of the N-terminal leucine-rich repeat region, which stems from using an alternative splice donor. Further research should reveal whether these differences underlie any distinct recruitment of CCR4–NOT complexes carrying these paralogs or whether apparent selectivity is determined by other factors, such as the length of the poly(A) tail, differential expression of the paralogs, or their specific localization in oocytes and zygotes.

Taken together, we show that loss of *Cnot6l* in mice results in reduced fertility. Although we cannot rule out that some of the effects observed in the oocyte or early embryos could be indirect effects of a role for *Cnot6l* (e.g., in granulosa cells), the phenotype is presumably a consequence of perturbed deadenylation and degradation of maternal mRNAs during OET. Because *Btg4*$^{-/-}$ eggs

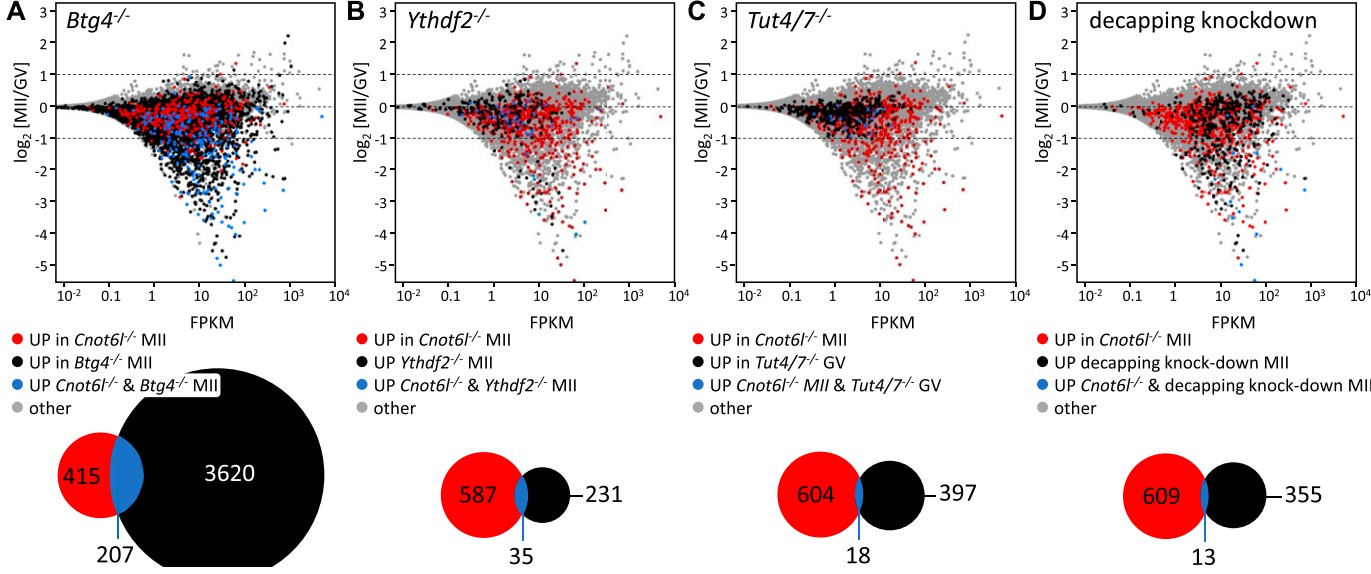

**Figure 5. Comparison of transcriptome changes in *Cnot6l*$^{-/-}$ MII eggs with other experimental data.**
**(A)** Comparison with *Btg4*$^{-/-}$ eggs (13). **(B)** Comparison with *Ythdf2*$^{-/-}$ eggs (15). **(C)** Comparison with *Tut4/7*$^{-/-}$ GV oocytes (28). **(D)** Comparison with eggs lacking production of the decapping complex (16). In each case, significantly up-regulated transcripts in MII eggs were compared with up-regulated transcripts in *Cnot6l*$^{-/-}$ MII eggs (Fig 3D). All MA plots were constructed from wild-type control replicates from GSE116771.

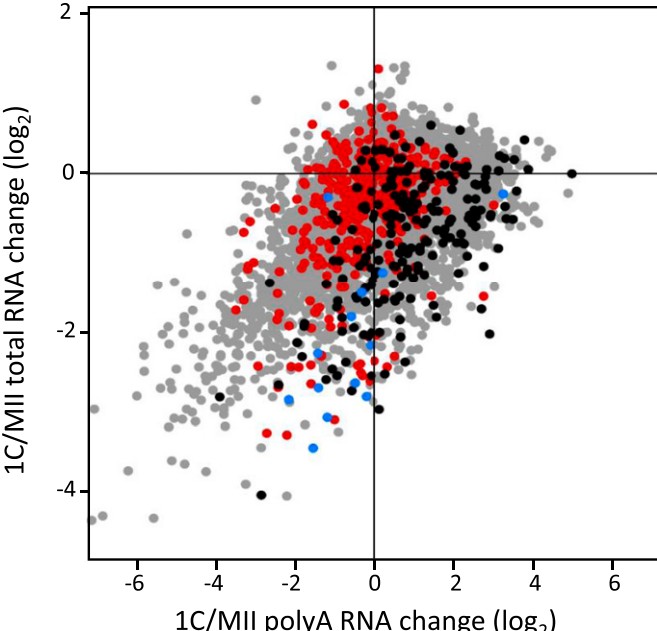

**Figure 6. Comparison of transcriptome changes in *Cnot6l*⁻/⁻ MII eggs with other experimental data.**
Projection of transcripts with relatively increased abundance in *Cnot6l*⁻/⁻ MII eggs and eggs with blocked decapping (Fig 5D) onto transcriptome changes in zygotes. The plot was constructed from published data (19, 20) as in Fig 4D. The *y*-axis shows relative changes in total RNA-seq (i.e., RNA degradation), whereas the *x*-axis shows poly(A) RNA changes from poly(A) RNA-seq (i.e., RNA degradation and/or deadenylation). In red are shown transcripts with significantly higher mRNA abundance in *Cnot6l*⁻/⁻ MII eggs. In black are shown transcripts with significantly higher mRNA abundance in MII eggs upon inhibition of decapping. In blue are shown transcripts up-regulated in both conditions.

exhibit much larger transcriptome changes than *Cnot6l*⁻/⁻ eggs, CNOT6L-mediated deadenylation appears rather selective. It is presently unclear if this selectivity stems from truly selective targeting (e.g., dependent on recruitment of the CNOT6L-containing CCR4–NOT complex directly through CNOT5L). Given a possible redundancy with the CAF1 component of the CNOT6L complex and/or other RNA degrading mechanisms, we speculate that a spectrum of transcripts targeted by the CNOT6L-containing CCR4–NOT complex is much broader and that transcripts showing a relative increase upon loss of *Cnot6l* are less targeted by redundant mechanisms.

# Materials and Methods

## Oocyte and embryo collection

Oocytes and early embryos were obtained from superovulated mice as described previously (29). Resumption of meiosis during culture

of GV oocytes was prevented with 0.2 mM 3-isobutyl-1-methylxanthine (Sigma-Aldrich). Animal experiments were approved by the Institutional Animal Use and Care Committees (approval no. 024/2012) and were carried out in accordance with the European Union regulations.

## Production of *Cnot6l* knockout model

*Cnot6l* knockout mice were produced by the Transgenic and Archiving Module of the Czech Centre for Phenogenomics (http://www.phenogenomics.cz/), Institute of Molecular Genetics ASCR, using TALENs (reviewed in reference 30) designed to delete coding exons 4–11 (Fig 2A). TALEN plasmids were produced as described previously (31). Target sites for TALEN pairs inducing an ~31.3-kb deletion (chr5: 96,075,067-96,106,332 [GRCm38/mm10]) were identified using the SAPTA tool (32) and TALEN off-targeting was addressed with the PROGNOS tool (33). The following RVD repeats were used to generate individual TALENs:

C6L-1R repeats NH NI HD NI NI NG NG NH HD NI NI NG NG NG HD—(cognate sequence: T0 GACAATTGCAATTTC)
C6L-1F repeats NG NI HD NG NI NG NH NG NG NH NG NG HD NG—(cognate sequence: T0 TACTATGTTGTTCT)
C6L-4F repeats NG NI NG NI NG NI HD HD NG HD NI NI NH HD NI NG HD HD—(cognate sequence: T0 TATATACCTCAAGCATCC)
C6L-4R repeats NH HD NH NG HD NI NH NH NG NI NI NG HD NI NI NG HD NG—(cognate sequence: T0 GCGTCAGGTAATCAATCT)

TALEN RNAs for injection were produced as described previously (31). A sample for microinjection was prepared by mixing all four TALEN RNAs in ultrapure water at a concentration of 4 ng/µl each. This mixture was loaded into the injection capillary and injected into male pronuclei of C57BL/6 one-cell embryos.

Genotyping was performed by PCR on lysates from tail biopsies from 4-wk-old animals using genotyping primers Cnot6l-1F (5′-GT-CATCAGGTTTGGCAGCAAGC-3′) and Cnot6l -1R (5′-CTAAGAAGTGTGTG-GTGCATCAGC-3′) for the wild-type allele (yielding a 597-bp product) and Cnot6l-1F and Cnot6l-2R2 (5′-CAGAGAAGAAAGCCCACCCG-3′) for the deletion (yielding a predicted 357-bp product).

## Analysis of preimplantation development

Mice were superovulated and cumulus oocyte complexes were isolated as described previously (29). Sperm of C57BL/6J (8–12 wk old) males were used for in vitro fertilization. Males were euthanized by cervical dislocation and sperm were isolated from cauda epididymis, capacitated for 1 h in human tubal fluid medium, and mixed with cumulus oocyte complexes in human tubal fluid. In vitro fertilization was performed for 5 h. Next, zygotes containing two pronuclei were selected and preimplantation development was analyzed by the PrimoVision time-lapse system (Vitrolife) with 15-min acquisition settings (t = 0; start of recording). Embryos were cultured in KSOM medium at 37°C under 5% $CO_2$ until the blastocyst stage. Automatically recorded times for preset cleavage events were confirmed by personal inspection and the median time was plotted against the embryonic stage. The experiment was repeated five times. Heterozygous and homozygous females used in each

experiment were littermates, the eggs were fertilized by a single male, and embryos were developed side-by-side in a single incubator. Representative recorded videos (Videos 1 and 2) are provided in the Supplementary Information.

### RNA-seq

Total RNA was extracted from triplicates of 25 wild-type or knockout GV oocytes, MII eggs, or one-cell zygotes using a PicoPure RNA isolation kit with on-column genomic DNA digestion according to the manufacturer's instructions (Thermo Fisher Scientific). Each sample was spiked in with 0.2 pg of synthesized *Renilla* luciferase mRNA before extraction as a normalization control. Non-stranded RNA-seq libraries were constructed using the Ovation RNA-seq system V2 (NuGEN) followed by the Ovation Ultralow Library system (DR Multiplex System; NuGEN). RNA-seq libraries were pooled and sequenced by 125-bp paired-end reading using the Illumina HiSeq at the High Throughput Genomics Core Facility at the University of Utah, Salt Lake City, UT. RNA-seq data were deposited in the Gene Expression Omnibus database under accession ID GSE116771.

### Bioinformatics analyses

#### *Mapping of Illumina RNA-seq reads on the mouse genome*
All RNA-seq data were mapped using the STAR mapper (34) version 2.5.3a as described previously (35), except all multi-mapping reads:

STAR −readFilesIn $FILE1 $FILE2 −genomeDir $GENOME_INDEX −runThreadN 8 −genomeLoad LoadAndRemove −limitBAMsortRAM 20000000000 −readFilesCommand unpigz −c −outFileNamePrefix $FILENAME −outSAMtype BAM SortedByCoordinate −outReadsUnmapped Fastx −outFilterMultimapNmax 99999 −outFilterMismatchNoverLmax 0.2 −sjdbScore 2.

The following genome versions were used for mapping the data: mouse—mm10/GRCm38, human—hg38/GRCh38, cow—bosTau8/UMD3.1, and hamster—MesAur1.0 (GCF_000349665.1).

Annotated gene models for all organisms corresponding to their respective genome versions were downloaded from the Ensembl database as gene transfer format files. Only protein-coding genes were used in all subsequent analyses. Data were visualized in the University of California, Santa Cruz (UCSC), genome browser by constructing bigWig tracks using the UCSC tools (36).

#### *Differential expression analysis of RNA-seq data*
Analysis of genes differentially expressed in knockouts compared with wild types in different developmental stages was performed in the R software environment. Mapped reads were counted over exons grouped by gene as follows.

GenomicAlignments::summarizeOverlaps(features = exons, reads = bamfiles, mode = "Union", singleEnd = FALSE, ignore.strand = TRUE).

Statistical significance and fold changes in gene expression were computed using the DESeq2 package (26) from RNA-seq data prepared as biological triplicates. Briefly, DESeq2 analysis starts with a matrix of read counts obtained with *summarizeOverlaps()* command above, in which each row represents one gene and each column one sample. Read counts are first scaled by a normalization factor to account for differences in sequencing depth between samples. Next, dispersion (i.e., the variability between replicates) is

calculated for each gene. Finally, negative binomial generalized linear model (GLM) is fitted for each gene using those estimates and normalized counts. GLM fit returns coefficients, indicating the overall expression strength of the gene and coefficients (i.e., $\log_2$-fold change) between treatment and control (in our analysis knockout and wild-type samples). Significance of coefficients in GLMs are tested with the Wald test. Obtained *P*-values are adjusted for multiple testing using the Benjamini and Hochberg False Discovery Rate procedure (37). In our analysis, expression changes with p-adjusted values smaller than 0.1 (the default DESeq2 cutoff) were considered significant.

#### *Differential expression analysis of microarray data (decapping complex, YTHDF2)*
Microarray data were normalized and background-corrected using RMA (38) (*Ythdf2* data) or GC-RMA (39) (decapping complex data) algorithms. Statistical significance and fold changes in gene expression were computed using SAM method (40).

#### *Principal component analysis plot*
Principal component analysis was computed on count data transformed using regularized logarithm (rlog) function from the DESeq2 (26) R package.

#### *Statistical analysis*
Fisher's exact test was used to evaluate the significance of number of genes showing increased transcript abundance in *Cnot6l*$^{−/−}$ MII eggs and MII eggs with reduced decapping complex or knockouts of *Ythdf2* or *Btg4*. A level of $P < 0.05$ was considered to be significant.

## Supplementary Information

## Acknowledgements

We thank Vedran Franke and Josef Pasulka for help with data analysis. This research was supported by the Czech Science Foundation (CSF) grant P305/12/G034 and by the Ministry of Education, Youth, and Sports (MEYS) project NPU1 LO1419. Additional support of coauthors included CSF grant 17-08605S to H Fulka; LM2015040 (Czech Centre for Phenogenomics), CZ.2.1.05/1.1.00/02.0109 (Biotechnology and Biomedicine Centre of the Academy of Sciences and Charles University), and CZ.1.05/2.1.00/19.0395 (higher quality and capacity for transgenic models) support by MEYS and RVO 68378050 by the Academy of Sciences of the Czech Republic to R Sedlacek; the European Structural and Investment Funds grant for the Croatian National Centre of Research Excellence in Personalized Healthcare (contract KK.01.1.1.01.0010), Croatian National Centre of Research Excellence for Data Science and Advanced Cooperative Systems (contract KK.01.1.1.01.0009), and Croatian Science Foundation (grant IP-2014-09-6400) to K Vlahovicek; and a grant from National Institutes of Health (HD022681) to RM Schultz.

### Author Contributions

F Horvat: conceptualization, data curation, software, formal analysis, supervision, funding acquisition, investigation, visualization,

project administration, and writing—original draft, review, and editing.

H Fulka: resources, formal analysis, investigation, and writing—review and editing.

R Jankele: resources, data curation, software, formal analysis, investigation, and writing—review and editing.

R Malik: conceptualization, funding acquisition, investigation, project administration, and writing—review and editing.

J Ma: data curation, software, formal analysis, investigation, and writing—review and editing.

K Solcova: investigation, methodology, and writing—review and editing.

R Sedlacek: resources, formal analysis, investigation, and writing—review and editing.

K Vlahovicek: resources, data curation, and software.

RM Schultz: conceptualization, funding acquisition, methodology, project administration, and writing—review and editing.

P Svoboda: conceptualization, formal analysis, supervision, funding acquisition, investigation, visualization, project administration, and writing—original draft, review, and editing.

## Conflict of Interest Statement

The authors declare that they have no conflict of interest.

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
