## [Reviewer comments · Life Science Alliance]

Role of *Cnot6l* in maternal mRNA turnover

Filip Horvat, Helena Fulka, Radek Jankele, Radek Malik, Ma Jun, Katerina Solcova, Radislav Sedlacek, Kristian Vlahovicek, Richard M. Schultz, and Petr Svoboda

DOI: 10.26508/lsa.201800084

Review timeline:

Submission Date:	5 May 2018
Editorial Decision:	11 June 2018
Revision Received:	3 July 2018
Editorial Decision:	4 July 2018
Accepted:	5 July 2018

Report:

(Note: Letters and reports are not edited. The original formatting of letters and referee reports may not be reflected in this compilation.)

1st Editorial Decision

11 June 2018

Thank you for submitting your manuscript entitled "Role of *Cnot6l* in maternal mRNA turnover" to Life Science Alliance. The manuscript was assessed by expert reviewers, whose comments are appended to this letter.

As you will see, all reviewers appreciate your analyses and provide constructive input on how to further strengthen your work in a minor revision.

I would thus like to invite you to submit a revised version of your manuscript, following the suggestions made by the reviewers. Mostly text changes / inclusion of available data are needed. However, the control for the knock-out efficiency requested by referee #3 that is mentioned in the M&M section but not shown, should be provided, ideally in conjunction with the western blots for residual/truncated *Cnot6l* expression.

Thank you for this interesting contribution to Life Science Alliance. We are looking forward to receiving your revised manuscript.

REFEREE REPORTS

Reviewer #1 (Comments to the Authors (Required)):

The authors explore the function of CNOT6L in oocyte to embryo transition (OET). The manuscript has three principal claims:

1. CNOT6L is the principal CNOT6 enzyme expressed in rodent oocytes.
2. CNOT6L is important for female mouse fertility and timing of early embryonic development.
3. CNOT6L is required for the metabolism of the maternal transcriptome but not the formation of maternal transcriptome.

All the principal claims are supported by the experimental evidence provided. The interpretation of the data is precise and reasonable. The core message of the manuscript is important and timely. I strongly support the manuscript for publication; however I do have a few minor comments & suggestions:

1. The authors should include 3'UTRs in the following statement: 'Control of mRNA stability involves both a 5' cap and a 3' poly(A) tail, structures targeted by different mRNA degradation

mechanisms (reviewed in [3]).!

2. The authors should highlight that CNOT6L is not required for the formation of the maternal transcriptome. This is a key point that is supported by the phenotypic and RNA-seq analysis. Also given that poly(A) length has been shown to be important in the formation of the maternal transcriptome (Morgan et al 2017). This could be discussed.

3. Complementary to point 2, the authors may want to determine the ratio of SN to NSN oocytes. This is not a formal requirement as this ratio is unlikely to be changed given the lack of phenotype in GV oocytes but it would merit a nice additional confirmation.

4. I think the use of NGS in the text should be substituted with RNA-seq.

5. Poly (A) tail length has been defined in GV oocytes (Morgan et al 2017). For the transcripts whose expression is changed during oocyte maturation in CNOT6L^{-/-} MII oocytes, do they already have short or shorter poly(A) tails in GV oocytes?

Reviewer #2 (Comments to the Authors (Required)):

In this manuscript from Horvat, et al., the authors explore the role of CNOT6L in mammalian oocyte maturation and embryogenesis. In mammals, maternal mRNAs deposited in the oocyte are cleared during maturation in a process that depends on deadenylation and decapping. One of the major deadenylase complexes is the CCR4-NOT complex, which, in mammals, can have one of four catalytically active proteins. Here, the authors describe a role for CNOT6L, which appears to be the major CCR4 component in oocytes. Mutant females have reduced fertility, probably because zygotes from these mothers were less likely to reach the blastocyst stage. The authors describe RNA-seq datasets from mutant oocytes (GV, MII and 1-cell stage) and characterize changes at the transcriptome-wide level that occur upon loss of CNOT6L. In general, I am enthusiastic about this manuscript, which will be of interest to those working in the OET/MZT field. However, I would like to see the following points addressed:

Major points

- 1) One of the major arguments supporting the role of CNOT6L is that its expression is higher than of CNOT6. What is the expression of these two genes in other tissues/developmental stages?
- 2) As the authors note, the RNA-seq libraries are influenced by poly(A) tail length, and so it is difficult for them to make absolute statements about the extent to which CNOT6L predominantly impacts poly(A) tail length (and thus, presumably, translation) or stimulates mRNA decay. Because of this issue, it would be very useful to look at several individual mRNAs by an alternative method- this would lend support for their model.
- 3) There are available datasets for mRNA tail length, uridylation, etc. It would be extremely valuable to ask how poly(A) tail lengths of CNOT6L targets change, whether they are shorter, tend to not be uridylated, etc.
- 4) Given that developmental phenotypes did not emerge until the 3-cell stage, it would be useful to extend the RNA-seq analysis to later time points (2- and 3-cell stages). It may be that the changes in poly(A) tail length or RNA levels are more pronounced at this point.

Minor points:

- 1) On pg 4, it is unclear what this sentence means: "The transition from mRNA stability to instability that accompanies oocyte maturation involves recruitment of dormant maternal mRNAs that encode components of mRNA degradation pathways."

Reviewer #3 (Comments to the Authors (Required)):

In this manuscript, the authors investigate the role of Cnot6l in female fertility. Maternal mRNA decay is an important component of cytoplasmic reprogramming across the oocyte-to-embryo transitions and mechanisms contributing to decay remain incompletely known. In this study, global knockout of Cnot6l resulted in reduced female fertility, with apparently normal oocyte development and decreased rates of embryo development and increased rates of embryo arrest. Effects on the

maternal mRNA pool were minimal at the GV stage but increased substantially at both the MII and zygote stages. That there was minimal overlap between maternal mRNAs downregulated by CNOT6L, DCP1/2, YTHDF2 and BTG4 is an interesting finding that opens the door for future studies. The study well-written, concise, carefully interpreted, scientifically solid overall, and confirms/expands on data in the field, thereby filling an important gap in the field. A few major and minor comments are detailed below.

Major comments:

1. Validation of Cnot6l knockout. Although the knockout was designed to remove a large portion of the Cnot6l coding region, validation of the knockout is incomplete. The authors show the sequence of presumed genomic PCR products in Fig 2B but the PCR itself for +/- and -/- mice used throughout is not shown. An absence of mRNA transcripts from the deleted region is shown in 3A; however, Fig. 4A shows a Cnot6l transcript clearly remains detectable by RNAseq. Knockout at the level of protein expression needs to be demonstrated by Western blotting to show the degree of knockout and whether or not a truncated version of the protein is present. Also, state what mouse strain was used for KO.
2. Fig. 5D - This figure is not completely clear. Are these data for mRNAs upregulated from GV to MII oocytes with DCP1/2 and CNOT6L KO overlaid on top of mRNAs normally upregulated during the transition from MII to 1C? If so, it is not clear to me how these data can be directly compared and more justification for the comparison and conclusions needs to be provided. If not, then the data used to construct the graph needs to be more explained more clearly.

Minor comments (line numbers would be helpful in the future):

1. Introduction, p.3 - First paragraph, last sentence - I would recommend making this statement a little less conclusive. To my knowledge, the sequential deadenylation mentioned has only been shown in one study/model and has not been demonstrated to be a universal mechanism applicable to all cell types, mRNAs or mRNA decay pathways.
2. Introduction, p.4 - I would recommend defining the concept of dormancy for a broad audience.
3. Introduction - The justification/scientific question for this study should be put in context with respect to what is known about role of CNOT6L from Ma et al, 2015 in the introduction.
4. Results - p.6 - SD should be given for litter sizes and p values for differences in litter sizes.
5. Results - p.6 - p value for difference in -/- and +/- zygote levels not given.
6. Results - p.6 - Did KO GV and MII oocytes look the same as controls? i.e., any abnormalities in polar body appearance, etc.?
7. Results - p.7 - 40% blast rate for controls is very low. Presumably this is because of IVF protocol but raises questions as to validity of a 2-fold difference. Would be better to show blast rates for in vivo fertilized embryos, not clear why this was not done.
8. Results - p.7 - Additional details as to differential expression analysis should be provided, including how expression was defined and what fold change and q value cutoffs were used to up- and downregulated transcripts.
9. Results - p.8 and Fig.3 - General comment to authors to be careful labeling up- and downregulated mRNAs as "stabilized" and "destabilized" given no half-life analyses were performed. Although it is fair to say this is the most likely explanation for changes in the majority of transcripts, it is likely residual transcriptional in the oocyte, transcription at minor ZGA, and/or differences in cytoplasmic polyadenylation/deadenylation (as they acknowledge) might also contribute to observed changes for some mRNAs.
10. Results - p.9 - Would be interesting information if the authors are able to meaningfully estimate what percentage of maternal mRNAs are downregulated by CNOT6L.
11. Results - p.9 - Would be interesting to know if is anything unique about the timing of decay for the subset of maternal mRNAs targeted by CNOT6L or if they have specific 3'UTR sequences, etc. Also, how does this subset overlap with mRNAs regulated by CNOT7? by MSY2? These would complete the comparisons if comparable data are available.
12. Discussion - There is no discussion of the fact that this is a global knockout. Therefore, it is possible some effects observed in the oocyte and early embryo could be indirect effects of a role for CNOT6L in granulosa or other cells. This should be acknowledged.

Reviewer #1

1. The authors should include 3'UTRs in the following statement: 'Control of mRNA stability involves both a 5' cap and a 3' poly(A) tail, structures targeted by different mRNA degradation mechanisms (reviewed in [3]).

The text has been modified to:

Control of mRNA stability involves various mechanisms, many employing protein interaction with the 3' untranslated region that ultimately targets the terminal 5' cap and 3' poly(A) tail structures (reviewed in [3])

2. The authors should highlight that CNOT6L is not required for the formation of the maternal transcriptome. This is a key point that is supported by the phenotypic and RNA-seq analysis. Also given that poly(A) length has been shown to be important in the formation of the maternal transcriptome (Morgan et al 2017). This could be discussed.

We revised the sentence describing the minimal changes in *Cnot6l*^{-/-} oocytes, to highlight the fact that *Cnot6l* is not required for the formation of the maternal transcriptome:

Analysis of differentially expressed transcripts using DESeq2 package [26] with the default p-value cut-off 0.1 showed minimal transcriptome changes in GV oocytes (only four transcripts showing a significant increase in abundance > 2-fold), which indicates that Cnot6l is not required for formation of the maternal transcriptome.

We included data analysis from (Morgan et al. 2017) in two ways:

- (1) We examined whether there is a relationship between maternal RNA poly(A) length and transcripts upregulated in *Cnot6l*^{-/-} oocytes during meiosis.
- (2) We examined whether the group of transcripts upregulated in *Tut4/Tut7* mutant oocytes overlaps with those upregulated in *Cnot6l*^{-/-} MII eggs. We did not include these data in the original manuscript as they were obtained from GV oocytes whereas we aimed at mechanisms affecting MII transcriptome composition. However, as it might be of interest of the readership that TUT4/7 targets seem to be apparently distinct from CNOT6L-regulated genes and decapping regulated genes, we included these data in Fig. 5C, which shows that *Ythdf2* and *Tut4/7* genes are less expressed than those sensitive to decapping.

3. Complementary to point 2, the authors may want to determine the ratio of SN to NSN oocytes. This is not a formal requirement as this ratio is unlikely to be changed given the lack of phenotype in GV oocytes but it would merit a nice additional confirmation.

Unfortunately, we did not analyze the chromatin configuration in the full-grown preovulatory oocytes as it seemed unimportant given the reported dormancy of the *Cnot6l* transcript and minimal transcriptome changes in preovulatory oocytes. Although we agree with the reviewer 1 that it would merit a nice confirmation, we think this is not an essential point that would require reviving the *Cnot6l*^{-/-} mouse strain.

4. I think the use of NGS in the text should be substituted with RNA-seq.

NGS was replaced with RNA-seq. We agree the term RNA-seq is more accurate.

5. Poly (A) tail length has been defined in GV oocytes (Morgan et al 2017). For the transcripts whose expression is changed during oocyte maturation in *CNOT6L*^{-/-} MII oocytes, do they already have short or shorter poly(A) tails in GV oocytes?

We added analysis of data from Tail-seq from (Morgan et al. 2017) – transcripts whose abundance increases during meiotic maturation of *Cnot6l*^{-/-} oocytes are having rather long poly(A) tails (60-80 nt). We added these results as new Figure 4B and 4C.

Reviewer #2

1) One of the major arguments supporting the role of CNOT6L is that its expression is higher than of CNOT6. What is the expression of these two genes in other tissues/developmental stages?

It appears that *Cnot6l* is less expressed in somatic tissues than *Cnot6*. We added an additional expression heatmap showing CCR4-NOT complex expression in 22 tissues from the ENCODE project.

2) As the authors note, the RNA-seq libraries are influenced by poly(A) tail length, and so it is difficult for them to make absolute statements about the extent to which CNOT6L predominantly impacts poly(A) tail length (and thus, presumably, translation) or stimulates mRNA decay. Because of this issue, it would be very useful to look at several individual mRNAs by an alternative method- this would lend support for their model.

We examined whether there is any relationship between poly(A) tail length and transcripts upregulated in *Cnot6l*^{-/-} mutant oocytes using poly(A) tail lengths of all maternal transcripts estimated by RNA-seq previously (Morgan et al. 2017). This analysis showed that *Cnot6l* does not predominantly impacts mRNAs with a specific poly(A) tail length (new Fig. 3B and 3C).

3) There are available datasets for mRNA tail length, uridylation, etc. It would be extremely valuable to ask how poly(A) tail lengths of CNOT6L targets change, whether they are shorter, tend to not be uridylated, etc.

As mentioned above, we included an additional data analysis concerning the relationship between polyA *Cnot6l*^{-/-} *Cnot6l*^{-/-} length in GV oocytes and maternal transcripts destabilized by Tut4/7 (by (Morgan et al. 2017)).

4) Given that developmental phenotypes did not emerge until the 3-cell stage, it would be useful to extend the RNA-seq analysis to later time points (2- and 3-cell stages). It may be that the changes in poly(A) tail length or RNA levels are more pronounced at this point.

The rationale for RNA-seq experiment design (i.e., to include GV, MII and 1-cell stages) was actually not to include the 2-cell stage because of the major wave of the zygotic transcription occurring at this stage. We targeted stages where post-transcriptional regulation is the dominant force shaping the transcriptome. The 2-cell transcriptome dynamics is complex because it involves major zygotic genome activation that is superimposed on formation of a transcriptionally repressive environment. Transcriptome changes in *Cnot6l* mutants at the 2-cell stage would combine post-transcriptionally affected maternal transcripts and consequent disturbance of transcriptional regulation, and hence would not offer a major advantage over the late 1-cell stage samples we analyzed.

Minor points:

1) On pg 4, it is unclear what this sentence means: "The transition from mRNA stability to instability that accompanies oocyte maturation involves recruitment of dormant maternal mRNAs that encode components of mRNA degradation pathways."

The text was revised and reads now:

Maternal mRNAs in mouse oocytes are unusually stable during the growth phase prior to oocyte maturation, which is accompanied with a transition from mRNA stability to instability (reviewed in [2]). This transition also involves recruitment of dormant maternal mRNAs that were accumulated but not (or poorly) translated during the growth phase. Dormant mRNAs encode components of mRNA degradation pathways [13,16,17] and include DCP1A and DCP2, which are critical components of the decapping complex [16].

Reviewer #3

Major comments:

1. Validation of *Cnot6l* knockout. Although the knockout was designed to remove a large portion of the *Cnot6l* coding region, validation of the knockout is incomplete. The authors show the sequence of presumed genomic PCR products in Fig 2B but the PCR itself for +/- and -/- mice used throughout is not shown. An absence of mRNA transcripts from the deleted region is shown in 3A;

however, Fig. 4A shows a *Cnot6l* transcript clearly remains detectable by RNAseq. Knockout at the level of protein expression needs to be demonstrated by Western blotting to show the degree of knockout and whether or not a truncated version of the protein is present. Also, state what mouse strain was used for KO.

Regarding validation of the knock-out, we include original sequencing chromatograms from analysis of PCR genotyping products of founder animals (new supplemental Figure S1). We also inspected individual reads mapping to the *Cnot6l* locus in knock-out animals to analyze splicing variants emerging upon the ~ 30 kb deletion (Figure S4):

- 1) We have found one read across the deletion, which confirms the original Sanger sequencing. Thus, the locus can only produce transcripts encoding only four N-terminal leucine rich repeats but not the deadenylase activity.
 - 2) We have identified in *Cnot6l* mutants splice variants of exon 4 splice donor with five different downstream splice acceptors (two splice acceptors were in intron 4, three downstream of the last exon). These are the same splice variants we have observed in mutants. Because the same variants would be present in heterozygotes, the risk is minimized that the phenotype caused by the deletion would stem from transcripts expressing a truncated CNOT6L protein. Please, note that the expression maximum depicted in Fig. S4 by a dashed line is 2 counts per million (CPM), whereas the maximum for normal *Cnot6l* expression was 50 CPM (Fig. 3A and S3)
- Regarding the validation of the knock-out by Western blot: We do not have a functional antibody recognizing the N-terminus of CNOT6L. Furthermore, another major obstacle is that the material for Western blot in oocytes is extremely scarce, particularly if one would like to detect proteins translated from low-level transcripts ~1 FPKM.

2. Fig. 5D - This figure is not completely clear. Are these data for mRNAs upregulated from GV to MII oocytes with DCP1/2 and CNOT6L KO overlaid on top of mRNAs normally upregulated during the transition from MII to 1C? If so, it is not clear to me how these data can be directly compared and more justification for the comparison and conclusions needs to be provided. If not, then the data used to construct the graph needs to be more explained more clearly.

We modified the text to explain better the figure design. The figure displays relative transcriptome changes between MII eggs and 1-cell zygotes resolved on two axes. On the y-axis, we show relative changes estimated from total RNA RNA-seq (Abe et al. 2015). This axis thus shows mRNA degradation independent of the poly(A) tail length changes. The x-axis shows relative changes in poly(A) RNA sequencing (RNA-seq of poly(A)-selected RNA (Karlic et al. 2017)). The point of the analysis is to show that transcripts upregulated in *Cnot6l*^{-/-} zygotes are distributed left relative to decapping-sensitive transcripts. A similar pattern is found when the x-axis is based on MII and 1-cell data from *Cnot6l* controls, which were based on total RNA sampling but reverse transcription included oligo dT priming (Fig. below). The main difference is that the poly(A)-selected dataset (left & figures in the manuscript) exhibits additional polyadenylation (orange rectangle), which is not apparent in *Cnot6l*^{+/-} controls (right panel). However, the shift to the left of the *Cnot6l*^{-/-} upregulated genes (red points) is still apparent.

Although we are aware that different high-throughput expression analysis platforms and even different RNA-seq experiments cannot be compared directly, it is valid to compare relative changes (Bottomly et al. 2011), which was done in this case.

Minor comments

1. Introduction, p.3 - First paragraph, last sentence - I would recommend making this statement a little less conclusive. To my knowledge, the sequential deadenylation mentioned has only been shown in one

study/model and has not been demonstrated to be a universal mechanism applicable to all cell types, mRNAs or mRNA decay pathways.

Introduction was revised, also to include the two most recent CCR4-NOT papers.

2. Introduction, p.4 - I would recommend defining the concept of dormancy for a broad audience.

There was a revision in response to the reviewer 1, which also incorporates a brief characterization of dormancy. The revised text reads:

Maternal mRNAs in mouse oocytes are unusually stable during the growth phase until oocyte maturation, which is accompanied with the transition from mRNA stability to instability (reviewed in Svoboda et al., 2015). This transition involves recruitment of dormant maternal mRNAs that accumulated without being translated during the growth phase. Dormant mRNAs encode components of mRNA degradation pathways (Ma et al., 2013, Ma et al., 2015, and Yu et al., 2016) and include DCP1A and DCP2, which are critical components of the decapping complex (Ma et al., 2013).

3. Introduction - The justification/scientific question for this study should be put in context with respect to what is known about role of CNOT6L from Ma et al, 2015 in the introduction.

We used a neutral transition into the last paragraph without a specific justification/scientific question with respect to Ma et al., 2015 (Ma et al. 2015) because the work on *Cnot6l* in the presented manuscript started before the analysis of *Cnot7* and *Cnot6l* dormancy, which was published as a separate manuscript. The neutral transition seems to us a better solution than making up a justification/scientific question based on data, which would fit into the Introduction but actually did not exist when we started the *Cnot6l* project.

4. Results - p.6 - SD should be given for litter sizes and p values for differences in litter sizes.

We revised the breeding table and included standard deviations to litter sizes, p-values for comparisons with litter sizes produced by heterozygote matings are mentioned in the text (p-values <0.01). Please, note in the first three rows litter sizes changed because pups from two litters recorded under the same cage number were accidentally counted as one litter.

5. Results - p.6 - p value for difference in -/- and +/- zygote levels not given.

We revised the sentence and added a p-value from Fisher's test of the difference of egg fertilization: *Accordingly, we analyzed early development of zygotes derived from Cnot6l^{-/-} and Cnot6l^{+/-} eggs fertilized in vitro with wild-type sperm and observed a small but significant reduction of fertilization efficiency of Cnot6l^{-/-} vs. Cnot6l^{+/-} eggs (85 vs. 99%; 133/155 Cnot6l^{-/-} vs. 203/205 Cnot6l^{+/-} eggs formed zygotes; Fisher's test p-value <0.001).*

We also added a comment on statistical analysis into Fig. 2C data where time differences were significant at all stages for both egg genotypes at the p-value cut-off <0.001. (in the first submission, we reported higher statistical significance of the difference for stages from 3-cell on (t-test, p-value <0.0001)).

6. Results - p.6 - Did KO GV and MII oocytes look the same as controls? i.e., any abnormalities in polar body appearance, etc.?

GV oocytes and MII eggs appeared normal (same as controls) but we did not perform any systematic analysis regarding polar body size, spindle abnormalities, or aneuploidy.

7. Results - p.7 - 40% blast rate for controls is very low. Presumably this is because of IVF protocol but raises questions as to validity of a 2-fold difference. Would be better to show blast rates for in vivo fertilized embryos, not clear why this was not done.

This seems to be a misunderstanding: 20% of the controls failed to develop to the blastocyst, i.e. 80% of the controls developed to the blastocyst. At the same time, 40% of the mutants failed to develop, i.e. 60 of the mutants did develop to the blastocyst stage. We modified the statement to *Importantly, although there was no stage-specific arrest of development for Cnot6l^{-/-} fertilized eggs, they were two times more likely to fail to reach the blastocyst stage than their Cnot6l^{+/-}-derived counterparts (38/95 (40.0%) Cnot6l^{-/-} vs. 26/133 (19.6%) Cnot6l^{+/-} fertilized eggs failed to develop to the blastocyst).*

8. Results - p.7 - Additional details as to differential expression analysis should be provided, including how expression was defined and what fold change and q value cutoffs were used to up- and downregulated transcripts.

We revised the Method section to explain better the differential expression analysis. We included in the analysis all significantly changed transcripts based on the default DESeq2 package p-value cut-off 0.1. We did not employ fold-change cut-offs as the expression changes were generally relatively small.

9. Results - p.8 and Fig.3 - General comment to authors to be careful labeling up- and downregulated mRNAs as "stabilized" and "destabilized" given no half-life analyses were performed. Although it is fair to say this is the most likely explanation for changes in the majority of transcripts, it is likely residual transcriptional in the oocyte, transcription at minor ZGA, and/or differences in cytoplasmic polyadenylation/deadenylation (as they acknowledge) might also contribute to observed changes for some mRNAs.

We revised the text, which reads now:

Interestingly, the numbers of significantly upregulated and downregulated mRNAs were comparable (Fig. 3E), which was unexpected because transcript stabilization would be the primary expected effect of a deadenylase component loss from the CCR4-NOT complex.

10. Results - p.9 - Would be interesting information if the authors are able to meaningfully estimate what percentage of maternal mRNAs are downregulated by CNOT6L.

We could estimate the percentage of maternal transcripts only from transcripts per million (TPM) values. The upregulated transcripts in *Cnot6l^{-/-}* MII eggs would concern 1.28 percent of the maternal transcriptome. However, without further experimental assessment, it's questionable how meaningful this value (which can be independently estimated from the released RNA-seq data) actually is because it based on relative transcriptome changes and does not reflect absolute poly(A) RNA changes. In any case, we added a comment to the text the following sentence:

The magnitude of transcriptome disturbance appears small despite the number of significantly affected genes; if RNA-seq data would be quantified as transcripts per million, higher transcript levels of 622 genes in MII eggs (Fig. 3E) would account for 1.28% of the transcriptome.

11. Results - p.9 - Would be interesting to know if is anything unique about the timing of decay for the subset of maternal mRNAs targeted by CNOT6L or if they have specific 3'UTR sequences, etc. Also, how does this subset overlap with mRNAs regulated by CNOT7? by MSY2? These would complete the comparisons if comparable data are available.

We have no data concerning the timing of decay of the maternal mRNAs targeted by CNOT6L. Regarding the motif analysis, we did not see any specific motifs globally associated with maternal mRNA destabilization during meiosis previously. It appears that the first wave of mRNA

degradation is relatively promiscuous whereas specific 3'UTR motifs are associated with mRNA stabilization (discussed in (Svoboda et al. 2017)). Unfortunately, transcriptome data from *Cnot7* knock-down are not available, the analysis employed qPCR of selected transcripts. We considered including *Msy2* knock-out transcriptome analysis but decided not to include it. The transcriptome in *Msy2* knock-out oocytes is globally disturbed already in full-grown GV oocytes, i.e., before meiotic maturation. Thus, it would be problematic to interpret an overlap with transcripts regulated in CNOT6L-dependent manner during meiotic maturation. In the revised version, we added data from (Morgan et al. 2017), which include transcriptome analysis of Tut4/7 knock-out oocytes and poly(A) tail length estimation by Tail-seq.

12. Discussion - There is no discussion of the fact that this is a global knockout. Therefore, it is possible some effects observed in the oocyte and early embryo could be indirect effects of a role for CNOT6L in granulosa or other cells. This should be acknowledged.

The revised version is addressing this issue in the statement:

Taken together, we show that loss of Cnot6l in mice results in reduced fertility. Although we cannot rule out that some of the effects observed in the oocyte or early embryos could be indirect effects of a role for Cnot6l (e.g., in granulosa cells), the phenotype is presumably a consequence of perturbed deadenylation and degradation of maternal mRNAs during OET.

REFERENCES

- Abe K, Yamamoto R, Franke V, Cao M, Suzuki Y, Vlahovicek K, Svoboda P, Schultz RM, Aoki F. 2015. The first murine zygotic transcription is promiscuous and uncoupled from splicing and 3' processing. *EMBO J* **34**: 1523-1537.
- Bottomly D, Walter NA, Hunter JE, Darakjian P, Kawane S, Buck KJ, Searles RP, Mooney M, McWeeney SK, Hitzemann R. 2011. Evaluating gene expression in C57BL/6J and DBA/2J mouse striatum using RNA-Seq and microarrays. *PLoS One* **6**: e17820.
- Karlic R, Ganesh S, Franke V, Svobodova E, Urbanova J, Suzuki Y, Aoki F, Vlahovicek K, Svoboda P. 2017. Long non-coding RNA exchange during the oocyte-to-embryo transition in mice. *DNA Res* **24**: 129-141.
- Ma J, Fukuda Y, Schultz RM. 2015. Mobilization of Dormant Cnot7 mRNA Promotes Deadenylation of Maternal Transcripts During Mouse Oocyte Maturation. *Biology of reproduction* **93**: 48.
- Morgan M, Much C, DiGiacomo M, Azzi C, Ivanova I, Vitsios DM, Pistolic J, Collier P, Moreira PN, Benes V et al. 2017. mRNA 3' uridylation and poly(A) tail length sculpt the mammalian maternal transcriptome. *Nature* **548**: 347-351.
- Svoboda P, Fulka H, Malik R. 2017. Clearance of Parental Products. *Adv Exp Med Biol* **953**: 489-535.

Thank you for submitting your revised manuscript entitled "Role of Cnot6l in maternal mRNA turnover". I appreciate the introduced changes, and I am happy to accept your manuscript in principle for publication in Life Science Alliance. Congratulations on this nice work!